# IL-18 favors Th2 responses in sporotrichosis caused by *Sporothrix globosa*, prolonging the course of the disease

Henan Si⊙, Yan Cui, Jianjiao Zu, Ying Shi, Yang Song, Yu Zhen⊙⊚*, Shanshan Li⊚*

Department of Dermatology and Venerology, the First Hospital of Jilin University, Changchun, China

⊚ These authors contributed equally to this work.
* zheny2016@jlu.edu.cn (YZ); lishans@jlu.edu.cn (SL)

## Abstract

### Background

Sporotrichosis is a subcutaneous mycosis caused by members of the genus *Sporothrix*, among which *Sporothrix globosa* (*S. globosa*) is the predominant etiological agent in Asia. T cell immunity plays crucial roles in fungal infections, yet the local T cell immune profile in sporotrichosis lesions remains unclear. IL-18, a pleiotropic cytokine capable of modulating T cell responses, is also poorly understood in host defense against *S. globosa*.

### Methodology/principal findings

qPCR, western blot, and IHC/mIHC were employed to profile IL-18, IL-18 BP, caspase-1 and IL-18R axis, along with Th1,Th2, and Th17 cells and their specific cytokines in sporotrichosis lesions versus healthy skin. Cellular sources of IL-18 in the lesions were identified via mIHC, and IL-18 production from keratinocyte/PBMCs exposed to *S. globosa* in vitro was tested by qPCR and western blot. Flow cytometry was performed to evaluate the role of IL-18 in Th2 polarization in the PBMCs-*S. globosa* coculture system. Lesional skin exhibited hyperactivated IL-18 signaling, marked by upregulated IL-18, caspase-1, and IL-18R, alongside decreased IL-18 BP. IL-18 was primarily released by dermal dendritic cells and Langerhans cells, rather than keratinocytes. A mixed Th1/Th2/Th17 microenvironment with Th2 predominance correlated strongly with prolonged disease duration. Strikingly, IL-18 synergistically interacted with Th1-derived IL-2 to establish a self-reinforcing Th2 loop, as evidenced by the strong correlation between lesional IL-4 and IL-18 levels ($r = 0.70$) and the concomitant upregulation of IL-18/IL-2 during *S. globosa*-induced Th2 expansion in PBMCs—an effect reversed by IL-18/IL-2 neutralizing antibodies. Critically, this Th2 skewing was mechanistically dependent on NF-κB signaling, as demonstrated through pharmacological pathway inhibition.

**Data availability statement:** The authors confirm that all data underlying the findings are fully available without restriction. All relevant data are within the paper and its Supporting Information files.

**Funding:** This work was supported by the National Natural Science Foundation of China (NO. 82373492 to SL), the Special Project for Medical and Health Talents of Jilin Provincial Department of Finance (NO. JLSWSRCZX2025-096 to YZ) and the Natural science Foundation of Jilin Province (NO. YDZJ202201ZYTS030 to YS). The funders had no role in study design, data collection and analysis, decision to publish, or preparation of the manuscript.

**Competing interests:** The authors have declared that no competing interests exist.

## Conclusions/significance

This study unveils the dual role of IL-18 in human sporotrichosis caused by *S. globosa*—amplifying both Th1 and Th2 responses but ultimately driving pathogenic Th2 polarization through IL-2 crosstalk. Our work identifies IL-18/IL-2/NF-κB axis as a key Th2-polarizing mechanism driving chronicity in this disease. Targeting this axis could recalibrate anti-fungal immunity, offering translational strategies for this subcutaneous disease.

## Author summary

Sporotrichosis, a subcutaneous fungal infection, has recently been designated as a neglected tropical disease by the World Health Organization, due to its rising global incidence with insufficient research and public health attention. *S. globosa* is one of the causative agents with a worldwide distribution, prevalent in Asia. However, knowledge about host immunity against this fungus remains limited. IL-18 is an inflammatory cytokine that can regulate T cell immunity within its cytokine milieu. Our study aims to characterize the T cell immune profile within skin lesions of sporotrichosis patients and to elucidate the role of IL-18 in modulating local immune responses against *S. globosa* infection, as well as its impact on disease progression. Our findings reveal a previously unrecognized predominant Th2-mediated immune response in the pathogenesis of sporotrichosis caused by *S. globosa*. This Th2 immunity is facilitated by IL-18 secreted by DCs and LCs, together with Th1-derived IL-2, and leads to prolonged disease duration in patients. Our study significantly advances the understanding of regulatory effects of IL-18 on T cell-mediated immunity and the pathogenesis of sporotrichosis.

## Introduction

Sporotrichosis is a subacute or chronic subcutaneous mycosis caused by species of *Sporothrix* complex [1], among which *Sporothrix schenckii* (*S. schenckii*), *Sporothrix brasiliensis* (*S. brasiliensis*), and *S. globosa* are the most clinically relevant. *S. globosa* has worldwide distribution with predominance in Asia, notably in northeast China [2–5], and causes milder symptoms due to its lower virulence. Comparative research shows that these three species are differentially recognized by human antigen-presenting cells [6], suggesting that *S. globosa* may elicit distinct immune responses in the host.

*Sporothrix* infection is typically localized to the skin, as systemic forms are rare in clinic except in immunocompromised individuals. Thus, local skin immunity plays a pivotal role to disease progression. However, up to now, aside from the infiltration of macrophages, neutrophils and T cells [7–9], little is known about the detailed immune profile of skin lesions in patients. Moreover, current

research about anti-*Sporothrix* immunity has largely focused on *S. schenckii* and *S. brasiliensis*, using murine models and in vitro cellular studies [6,10], leaving many details about immune responses to *S. globosa* in patients poorly understood.

Mouse models have highlighted the importance of T cell immunity in the defense against this fungus [11]: CD4[+] T cells suppress *S. schenckii* growth in vitro [12]; Th1 response was found throughout the immunity against *S. schenckii* infection and Th1/Th17 response was essential for optimal clearance of *S. schenckii* and *S. brasiliensis* [10,11,13–16], while Th2 response was related to the later phase of *S. schenckii* infection [17]. Previously, we revealed Th2-biased immunity in the circulation of sporotrichosis patients infected with *S. globosa* [18], different from the responses induced by the other two species in mice. However, whether the local immunity in skin lesions is consistent with the change we found in circulation and its potential regulatory factors is still unexplored.

Cytokines in the infection microenvironment can critically regulate T-cell immunity [19,20]. IL-1-dependent Th1 responses are essential against *S. brasiliensis* [21]. IL-18, an IL-1 family cytokine, can shape Th1, Th2, and Th17 responses depending on the cytokine milieu, specifically IL-12, IL-2 and IL-23 respectively [20,22–29]. It drives protective Th1 responses via IFN-γ induction in bacterial (e.g., *Yersinia enterocolitica*, *Mycobacterium tuberculosis*) and viral (e.g., *mouse hepatitis virus*) infections [21,30–32]. Similar effects of IL-18 have been reported in fungal infections caused by *Aspergillus*, *Cryptococcus neoformans*, *Candida* and *Paracoccidioides brasiliensis* [21,33–36]. Contrastingly, in *Leishmania*-infected murine models, IL-18 exacerbates disease by amplifying Th2-associated IL-4 production [37–39]. Nevertheless, the immunological role of IL-18 in human *Sporothrix* infection, particularly in determining Th1/Th2 polarization, remains poorly understood. Existing evidence is limited to two murine studies showing *S. schenckii*-infected splenocytes producing IL-18 *in vitro* upon stimulation with fungal components [40,41]. Thus, there remains a gap in our understanding regarding the detailed roles of IL-18 in sporotrichosis patients, especially caused by *S. globosa* infection.

To elucidate these questions, we investigated the in situ immune profile of skin lesions of patients and attempted to figure out the modulatory role of IL-18 in T cell responses against *S. globosa* infection and its impact on the progression of sporotrichosis.

## Materials and methods

### Ethics statement

The study was approved by the First Hospital of Jilin University Research Ethics Committee (NO. 24K077-001). In line with the Declaration of Helsinki, a written informed consent was obtained from each participant prior to study inclusion.

### Subjects

A total of 44 patients diagnosed with sporotrichosis (SP) and 40 age- and sex-matched healthy controls (HCs) were recruited in this study. None of the patients had underlying systemic diseases or were on immunosuppressive treatment. Isolates from patients were identified as *S. globosa* based on morphological characteristics and calmodulin-encoding gene (CAL) sequencing. Patients were divided into subgroups by disease duration (till diagnosis) and clinical types: fixed cutaneous form (FF) and lymphocutaneous form (LF); shorter duration (SD, < 6 months) and longer duration (LD, > 6 months). Clinical data and detailed demographics of the patients were listed in S1 Table.

### Skin and peripheral blood

Skin samples from healthy adults were obtained as surgical discards from cutaneous surgeries performed primarily for cosmetic purposes. Sporotrichosis skin specimens were collected via punch biopsies of patient lesions, while lymph nodes were obtained from residual surgical tissues. Peripheral blood samples from healthy controls were procured at the Physical Examination Center of the First Hospital of Jilin University.

## Strain and culture conditions

The *S. globosa* strain used was *Sporothrix globosa* ATCC4912. The yeast cells of *S. globosa* were obtained as previously described [18]. The yeast cells were heat-killed at 60°C for 2 h and then stored at 4°C.

## In vitro PBMCs culture

Peripheral blood mononuclear cells (PBMCs) isolated from healthy donors were co-cultured with *S. globosa* yeast cells at a multiplicity of infection (MOI) of 5, following established methodology [18]. For functional characterization, freshly isolated PBMCs ($2 \times 10^5$ cells/well) were plated in triplicate 96-well plates and exposed to yeast cells for 4 days under three treatment conditions: 1.2 μg/mL IL-18 antibody, 1 μg/mL IL-2 antibody, or 5 μM NF-κB pathway inhibitor BAY 11–7082 Parallel western blot experiments utilized $5 \times 10^6$ PBMCs/well in 6-well plates challenged with *S. globosa* (MOI = 5) for specified durations. To systematically evaluate IL-18-mediated Th2 polarization, PBMCs were stimulated with graded concentrations of recombinant human IL-18 (0, 50, 100 ng/mL) in the presence or absence of 10 ng/mL IL-2 over a 4-day culture period. Detailed information on antibodies and cytokines is provided in S2 Table.

## In vitro keratinocytes culture

Primary human keratinocytes were enzymatically isolated from discarded epidermal tissue obtained during dermatologic surgeries, as previously described [42]. Cells were expanded through two passages in T25 flasks using supplemented KGM medium (Lonza Bioscience, Basel, Switzerland) prior to experimental use. For fungal challenge studies, second-passage keratinocytes at 70% confluence in 6-well plates were exposed to *S. globosa* yeast cells (MOI = 5) for predetermined time intervals to assess IL-18 induction dynamics.

## Quantitative real-time PCR

qPCR was employed to characterize the mRNA expression of different genes. Detailed experimental methods are provided in S1 Appendix. The primer pair sequences used for qPCR are shown in S3 Table.

## Immunohistochemistry (IHC) and multiplexed immunohistochemistry (mIHC) assay

Formalin-fixed, paraffin-embedded tissue sections underwent IHC and mIHC, with quantitative histomorphometric analysis performed as described in the S1 Appendix, utilizing antibody specifications provided in S2 Table, and positive area quantification was conducted using ImageJ software.

## Flow cytometry

For detection of the cytokines, PBMCs were restimulated with Leukocyte Activation Cocktail for an additional 4 h. Then, cells were harvested and stained with surface and intracellular mAbs according to the manufacturer's instructions and analyzed with a flow cytometer. See S1 Appendix for more details.

## Western blot analysis

Briefly, protein extracts were separated by electrophoresis and transferred to a PVDF membrane. After blocking with 5% skimmed milk for 1.5 h, the membrane was incubated with primary antibodies overnight at 4°C (detailed information was listed in S2 Table). After subsequent incubation with HRP-conjugated secondary antibodies, the blots were assessed by enhanced chemiluminescence detection. Band intensities were quantified using ImageJ software with GAPDH serving as the loading control for normalization.

## Statistical analysis

Statistical analyses were performed using GraphPad Prism (v9.0.0, GraphPad Software) with unpaired Student's t-tests for two-group comparisons, one-way ANOVA for multi-group analyses, and Pearson's correlation coefficient tests to assess variable associations, where a two-tailed $P$ value < 0.05 defined statistical significance.

## Results

### IL-18 expression was upregulated in skin lesions of patients with sporotrichosis caused by *S. globosa*

As a member of alarmins, IL-18 plays important roles in host protection against bacterial, viral, and fungal infections [43]. To verify the effect of IL-18 on sporotrichosis caused by *S. globosa*, we first determined its local expression in lesional skin of patients. Compared with HCs, the mRNA levels of IL-18 and its receptor (IL-18R) were both significantly elevated in all biopsies of sporotrichosis patients (Fig 1A). Moreover, there were sharp increments of both molecules in the lesions of patients with long duration (LD), especially those with the lymphocutaneous form (LF). Similar changes were found in caspase-1 (S1A Fig), the lyase required for maturation of IL-18, while IL-18 binding protein (IL-18 BP), a high-affinity IL-18 decoy receptor and antagonist, was inversely reduced (Fig 1A). All these data indicated an enhanced IL-18 signaling in *S. globosa*-infected skin.

To further investigate IL-18 protein expression, we performed western blot analysis and quantified the protein levels using ImageJ software. IL-18 protein exists in two distinct forms: the inactive 24-kDa precursor (pro-IL-18) and the biologically active 18-kDa mature form. To distinguish between pro-IL-18 and mature IL-18, recombinant human IL-18 (18 kDa) was used as a molecular weight control. Remarkably, only a single 24-kDa band was detected in normal skin lysates, but in the lysates from sporotrichosis skin lesions, most of the immunoreactive IL-18 protein showed a molecular weight of 18 kDa, with a weak positive signal at the position of 24 kDa (Fig 1B). These findings indicate that only the precursor form of IL-18 is present in normal skin, whereas both the precursor and mature forms of IL-18 coexist in the sporotrichosis skin lesions, with the mature form being predominant. Consistent with the mRNA data, the mature IL-18 level was markedly elevated, particularly in lesions from patients with longer disease duration and lymphocutaneous form. In contrast, the expression level of pro-IL-18 in sporotrichosis skin lesions remained comparable to that observed in normal skin tissues (Fig 1B).

To investigate the distribution of IL-18 within sporotrichosis lesions, IHC staining was subsequently conducted. Unexpectedly, we found abundant IL-18 positive cells in the dermis of lesions (Fig 1C), especially in those from longer duration and lymphocutaneous form (Fig 1D). These findings were consistent with western blot data; while the distribution of epidermal IL-18-positive cells was comparable to that of HCs ($P = 0.2142$) with even lower expression intensity ($P < 0.0001$) (Fig 1E). Further correlation analysis showed a strong positive association between dermal IL-18 expression and disease duration ($P < 0.0001$, r = 0.89, Fig 1F). Similarly, the number of caspase-1-expressing cells was also markedly elevated in the dermis of lesions of all patients (S1B and S1C Fig), with no notable differences observed among lesions from patients with varying clinical types or disease durations (S1C and S1D Fig). However, both the distribution and expression intensity of caspase-1 in the epidermis of sporotrichosis lesions were lower than those found in HCs (S1E Fig).

### Keratinocytes were not the primary source of IL-18 in response to *S. globosa* infection

At the first line of skin barrier, human keratinocytes are capable of synthesizing and secreting proinflammatory factors, including IL-18, when exposed to various pathogen stimuli [23,44–47]. To clarify the ability of keratinocytes to produce IL-18 in response to *S. globosa* infection, we stimulated human primary keratinocytes with yeast cells of this fungus (5:1, yeast cells to keratinocytes, i.e., MOI = 5) in vitro and quantified mRNA/protein expression by qPCR/western blot at different time points. IL-18 mRNA was slightly increased after 6-hour incubation with *S. globosa* and reached the top level (less than 1.5-fold elevation) at 12 h and then returned to the baseline (Fig 2A), while there was no statistical change in protein

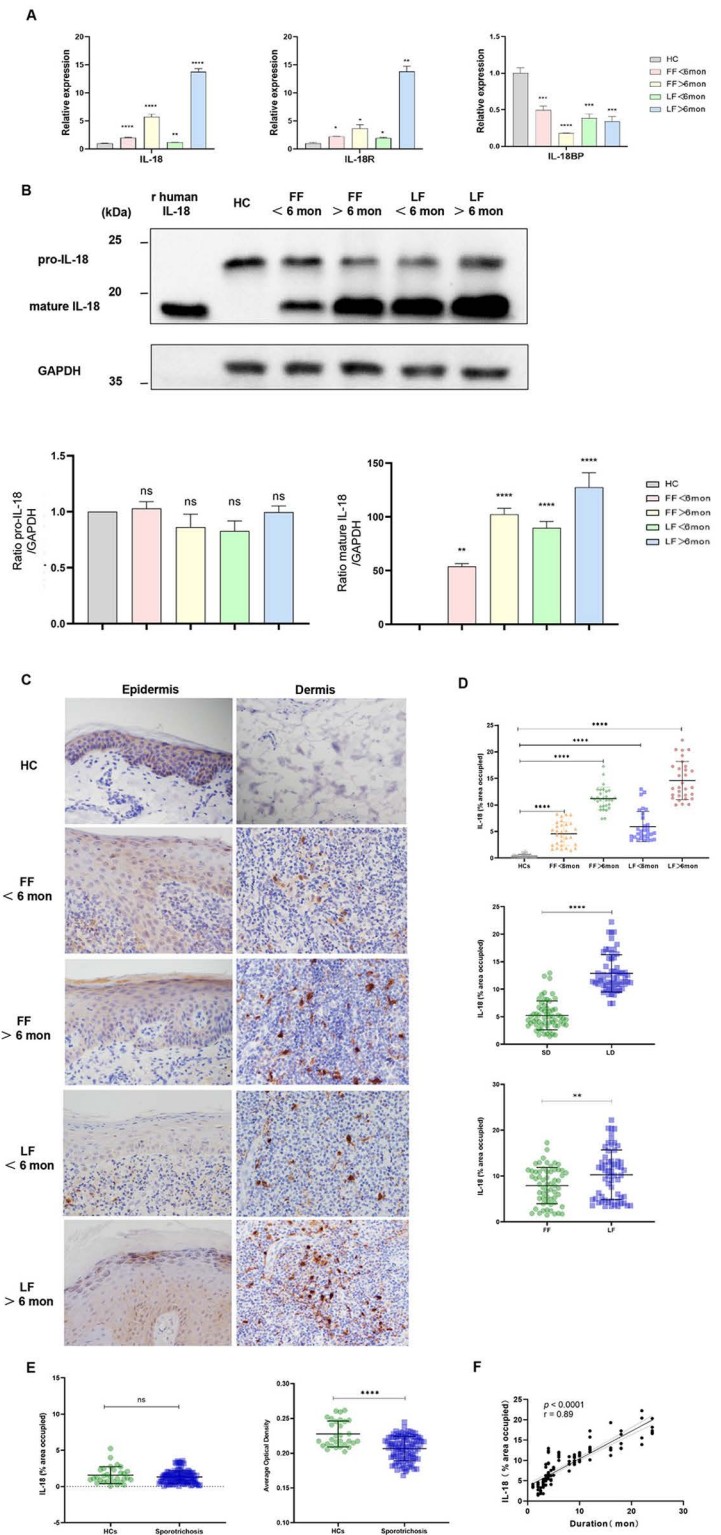

**Fig 1. IL-18 dysregulation in *S. globosa*-induced sporotrichosis skin lesions.** (A) qPCR analysis of IL-18, IL-18R, and IL-18 BP mRNA expression in HC skin versus sporotrichosis lesions. (B) Representative Western blot of IL-18 protein in HC and lesional skin (upper panel); densitometric quantification normalized to GAPDH (lower panel). (C) IHC staining of IL-18 in HC and lesional groups (400 × magnification). (D) Statistical analysis of IL-18

distribution area in the dermis of HC skins and sporotrichosis lesional skins (above); with comparisons between LD and SD (middle); between FF and LF (below). (E) Epidermal IL-18 distribution area and average optical density (AOD) in HC vs lesions. (F) Correlation between dermal IL-18 expression and disease duration. Sample sizes: HCs: n = 6, lesions: n = 24, n = 6 per subgroup. The data shown are expressed as mean ± SD of three independent experiments. Individual data points overlaid. Dashed lines in (F) indicate 95% confidence interval. Pearson's correlation coefficient (r) and associated $P$ values are shown on the graph. *$P$ < 0.05, **$P$ < 0.01, ***$P$ < 0.001, ****$P$ < 0.0001, and ns $P$ > 0.05. Abbreviations: HC: healthy control; FF: fixed cutaneous form; LF: lymphocutaneous form; SD: shorter duration; LD: longer duration; Mon, month.

expression as shown in Fig 2B. In addition, the bands derived from keratinocyte extracts were at the position of 24 kDa (Fig 2B), distinctly above the recombinant human IL-18 band (18 kDa) (S2 Fig), which was consistent with the molecular weight of the precursor form. These findings demonstrate that keratinocytes constitutively secrete pro-IL-18, suggesting that they are unlikely to contribute significantly to the pool of bioactive IL-18 in the skin during *S. globosa* infection.

### PBMCs were capable of secreting activated IL-18 when exposed to *S. globosa* in vitro

While prior studies indicate lymphocyte-derived IL-18 [29,48], our analysis revealed that most of IL-18-expressing cells in sporotrichosis dermis exhibited atypical lymphocyte morphology, displaying larger cell bodies and irregular dendritic processes (Fig 1B). To determine whether circulating lymphocytes contribute to IL-18 synthesis during *S. globosa* infection, we stimulated healthy donor-derived PBMCs with heat-killed fungal yeasts in vitro. Time-course analysis demonstrated rapid IL-18 mRNA induction, peaking at 2 h post-stimulation, then gradually decreased but remained statistically higher than control within 24 h (Fig 2C). The expression of IL-18R (Fig 2C) and caspase-1 (S3A Fig) was also elevated at mRNA levels in the initial 24 h. Correspondingly, IL-18 protein expression was significantly enhanced after 2 h incubation and reached its maximum at 6 h and declined at 24 h (Fig 2D); In contrast to human primary keratinocytes, IL-18 secreted by PBMCs was identified as an 18-kDa protein by western blot (Fig 2D), for its band aligns precisely with that of recombinant human IL-18 (S2 Fig). Moreover, cleaved caspase-1 increased at 4 h, peaked at 12 h and remained above the control level at 48 h (S3B Fig). Taken together, PBMCs displayed the ability to produce mature IL-18 when exposed to *S. globosa*.

### IL-18 was predominantly expressed by DCs and LCs in the lesions of patients with sporotrichosis caused by *S. globosa*

To identify the cells secreting IL-18 in lesional skin, we performed multiple IHC (mIHC) staining on biopsies from sporotrichosis patients and HCs. Irrespective of clinical type and duration, all skin lesions contained greatly increased CD11c+ dendritic cells (DCs), langerin+ Langerhans cells (LCs), CD68+ macrophages and CD3+ T cells in the dermis compared with healthy skin (Fig 3A and 3B). Moreover, most of IL-18+ cells were also CD11c+/langerin+, while only a small part of IL-18-producing cells were co-stained with CD68 and even fewer co-expressed CD3. Although murine LCs, macrophages, and T cells had been reported to produce IL-18 [25], our data showed that in human skin infected with *S. globosa*, DCs and LCs are the main IL-18-producing cells. Notably, we observed that IL-18 was distributed close to CD3+ T lymphocytes, suggesting a potential interaction between them.

### IL-18 was involved in Th cell responses in sporotrichosis lesional skin

We next investigated the potential role of IL-18 in modulating T cell responses during *S. globosa* infection. Our previous study showed a Th2-biased response in the circulation of patients with sporotrichosis caused by *S. globosa* [18]. To determine local changes in the lesions and explore the association between IL-18 and these Th cells, we initially evaluated the expression of CD4+IFN-γ+ Th1 cells, CD4+IL-4+ Th2 cells, and CD4+IL-17A+ Th17 cells in the tissues via mIHC staining and performed IHC staining for cytokines representative of Th1, Th2, and Th17 (i.e., IFN-γ, IL-4, and IL-17A) along with IL-18. Compared with HCs, Th1, Th2, and Th17 were all significantly increased in lesional skins ($P$ < 0.0001, Fig 4A and 4B), moreover, the distribution area of Th2 cells was much larger than those of Th1 and Th17 cells ($P$ < 0.0001, Fig 4B),

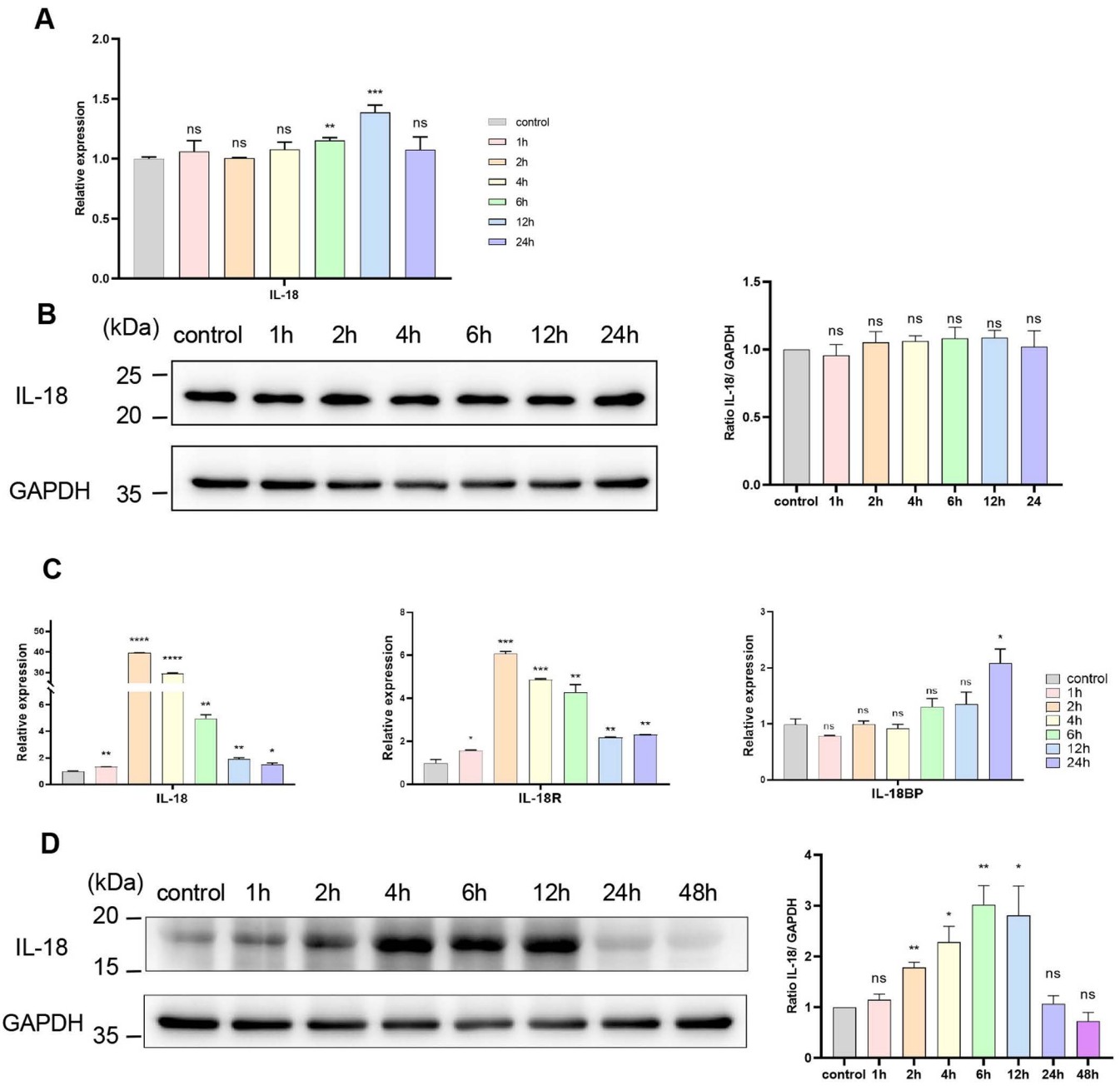

**Fig 2. The expression of IL-18 in human primary keratinocytes and PBMCs after exposure to *S. globosa* in vitro.** (A) The mRNA levels of IL-18 in keratinocytes post-*S. globosa* infection at different time points (MOI = 5; 0-24 hr). (B, D) Western blot analysis of IL-18 expression in keratinocytes (B) and PBMCs (D) after *S. globosa* stimulation (MOI = 5; left panel: representative blots; right panel: densitometric quantification normalized to GAPDH). (C) Temporal mRNA profiles of IL-18, IL-18R, and IL-18 BP in PBMCs following *S. globosa* challenge (MOI = 5; 0-48 hr). The data shown are expressed as mean ± SD of three independent experiments. *$P < 0.05$, **$P < 0.01$, ***$P < 0.001$, ****$P < 0.0001$, and ns $P > 0.05$.

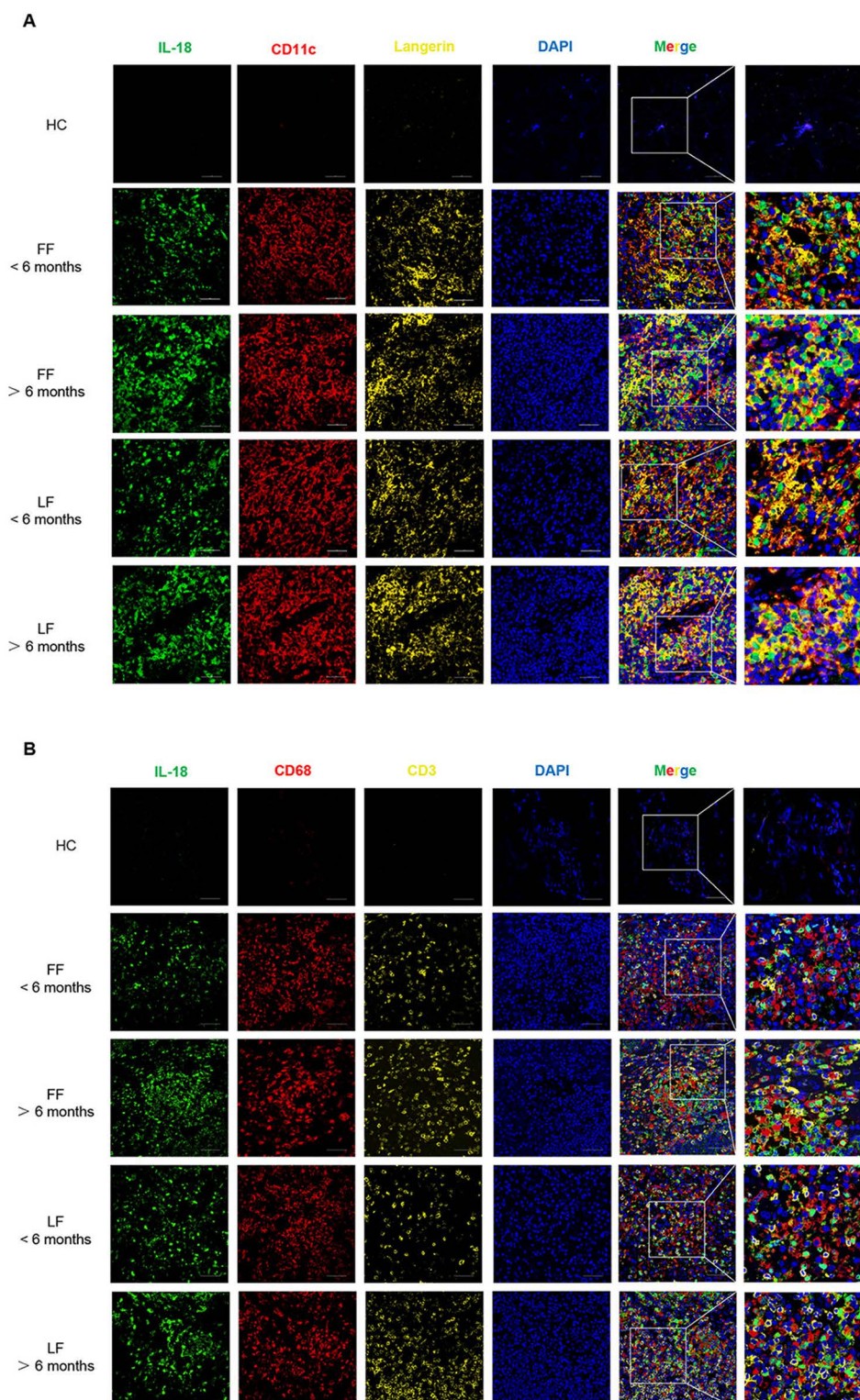

**Fig 3. Cellular sources of IL-18 in *S. globosa*-induced sporotrichosis lesions .** (A) Representative mIHC staining images of IL-18 (green), CD11c (red), Langerin (yellow), and 4′,6-diamidino-2-phenylindole (DAPI; blue) in each group. (B) Representative mIHC staining images of IL-18 (green), CD68 (red), CD3 (yellow), and DAPI (blue) in each group. Scale bars = 50 μm.

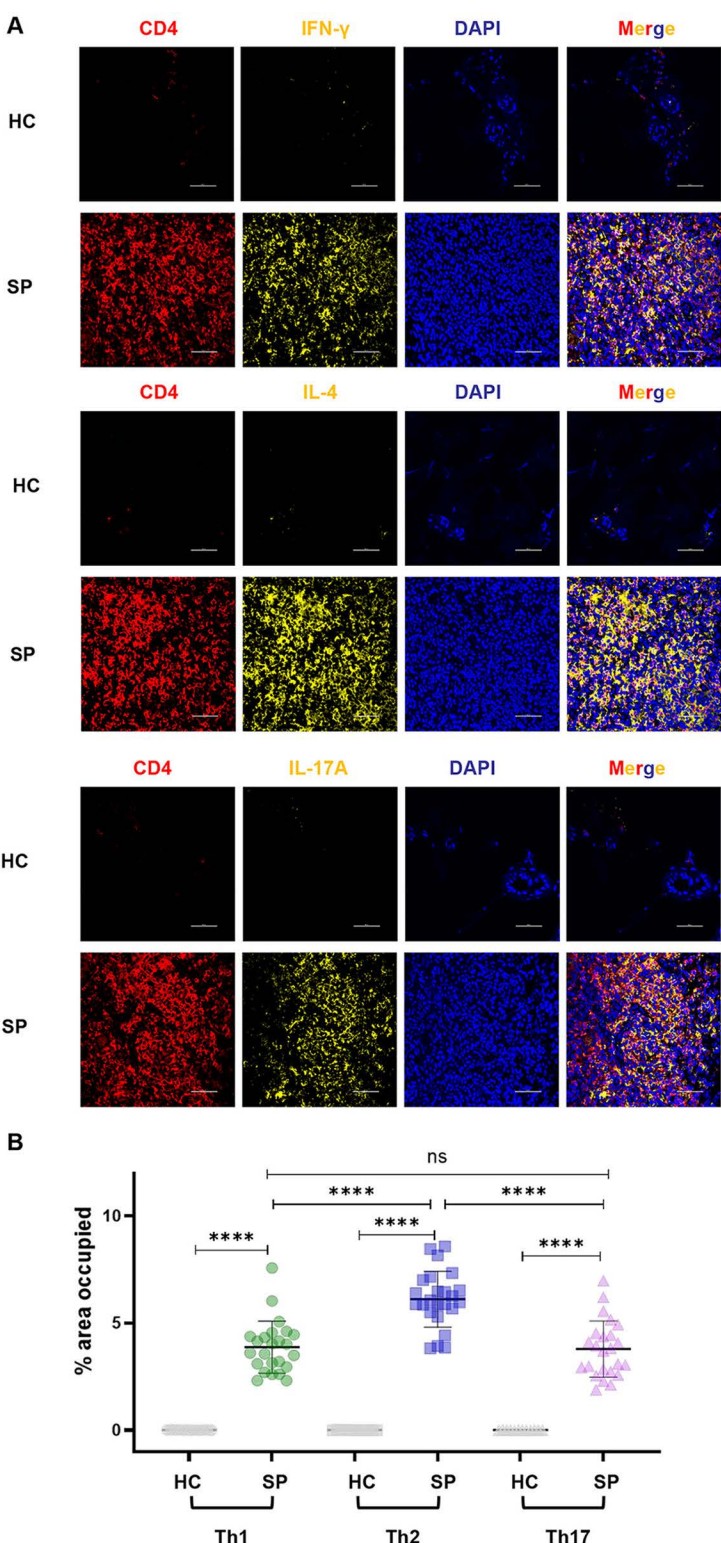

**Fig 4. Th cell profile in sporotrichosis lesional skin.** (A) mIHC visualization of Th subsets: CD4+ T cells (red) co-expressing IFN-γ (Th1, yellow), IL-4 (Th2, yellow), or IL-17A (Th17, yellow) in lesional vs healthy skin (DAPI, blue; scale bar = 50 μm). (B) Statistical analysis of Th1/Th2/Th17 infiltration in dermis. The measured values from individual patients were plotted by dots (HC: n = 4; SP: n = 8). ****$P < 0.0001$, and ns $P \geq 0.05$. SP, sporotrichosis·

Diseases

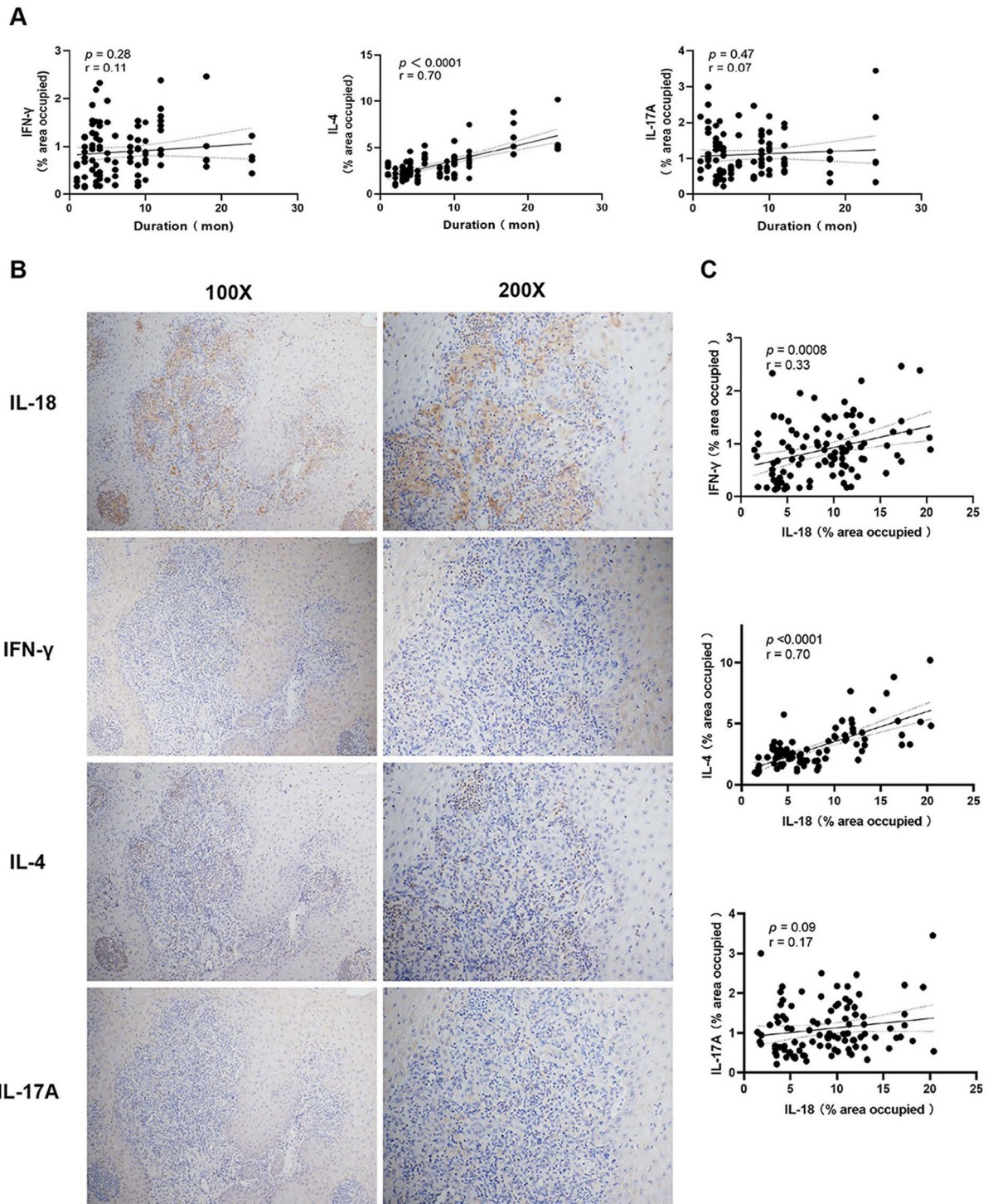

**Fig 5. The expression of IFN-γ, IL-4, and IL-17A in sporotrichosis lesional skin.** (A) Correlation analysis between IFN-γ, IL-4, and IL-17A distribution area in the dermis of sporotrichosis lesional skins and disease duration (n = 20). (B) Representative IHC images for IL-18, IFN-γ, IL-4, and IL-17A in the same sporotrichosis skin samples. (C) Correlation analysis of IL-18, IFN-γ, IL-4, and IL-17A expression (n = 20). The graphs showed a linear fit for patients. The data were analyzed with Pearson correlation analysis. The dotted line demonstrated a 95% confidence interval; r and associated *P* values are shown on the graph.

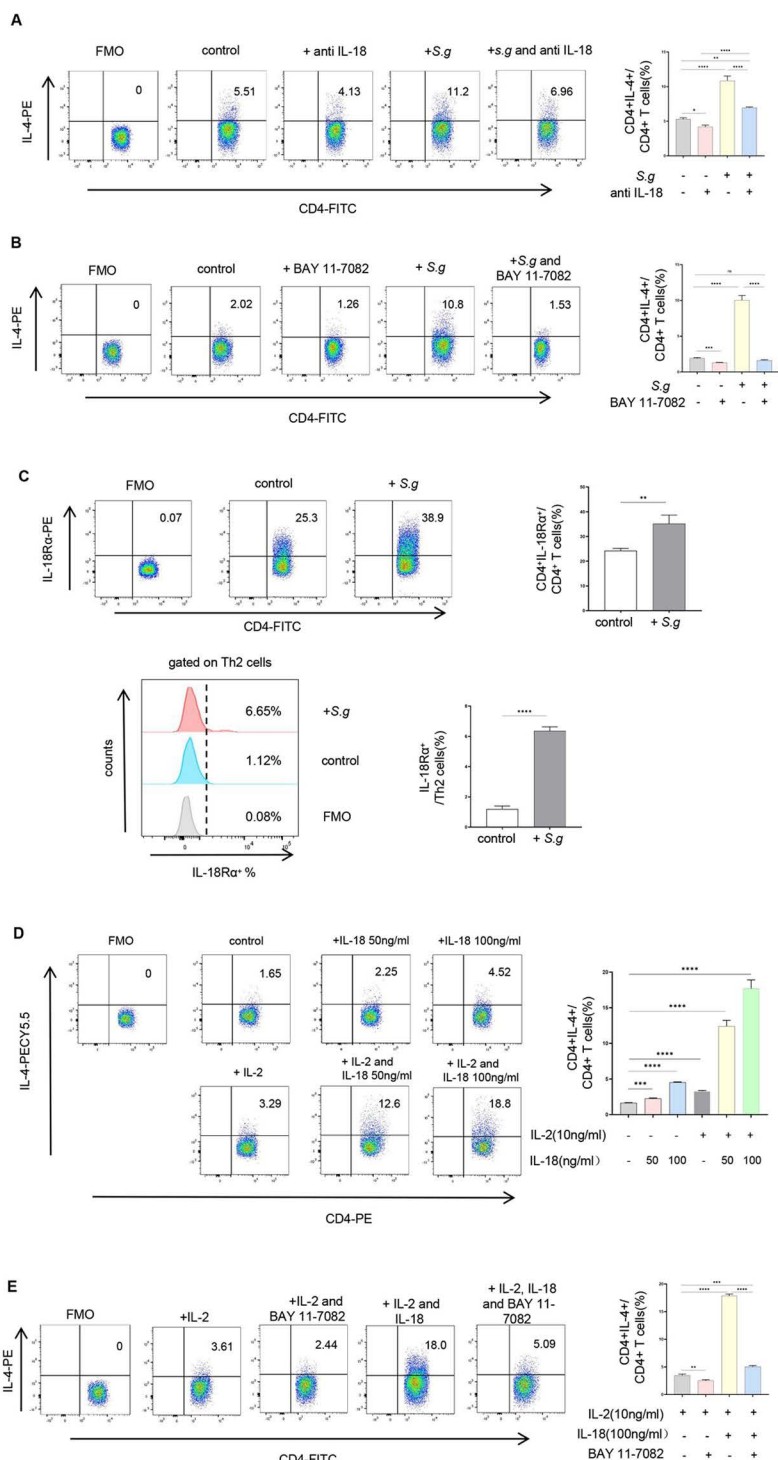

**Fig 6. The effect of IL-18 on Th2 induction in *S. globosa*-infected PBMCs in vitro.** (A, B) Effect of IL-18 neutralization (A) and NF-κB blockade (B) on *S. globosa*-induced CD4+IL-4+ Th2 cells in PBMCs. (C) The expression of IL-18Rα in CD4+ T cells and CD4+IL-4+ Th2 cells. (D) Impact of IL-18 alone or with IL-2 on the quantity of CD4+ IL-4+ Th2 cells. (E) Impact of blocking the NF-κB signaling pathway on the induction of IL-4 production by IL-18. Flow cytometry plots are shown on the left, with summary data displayed as bar graphs on the right. The data shown are expressed as mean±SD of three independent experiments. *P<0.05, **P<0.01, ***P<0.001, ****P<0.0001 and ns P≥0.05. *S.g.*, *Sporothrix globosa*. FMO, Fluorescence minus one.

indicating a stronger Th2 response in the lesions caused by *S. globosa*, which is consistent with the change we observed in the circulation [18]; moreover, akin to IL-18, only IL-4 exhibited a remarkable positive association with the duration of patients ($P < 0.0001$, r = 0.70; Fig 5A). Notably, IL-18-producing cells surrounded these Th cells (Fig 5B), and IL-18 had a strong positive correlation with the distribution of IL-4 ($P < 0.0001$, r = 0.70) and a moderate one with IFN-γ ($P = 0.0008$, r = 0.33) in all lesions regardless of clinical type and duration (Fig 5C), which strongly suggested the local interaction between IL-18 and Th2 plus Th1 cells. To verify this hypothesis, we cocultured PBMCs with heat-killed yeast cells in vitro for 4 days, and we found that the proportions of Th1, Th2, and Th17 in PBMCs were all elevated (Figs 6A and S4A). Meanwhile, anti-IL-18 antibody had no effect on Th17 expansion but decreased the frequencies of CD4+IL-4+Th2 cells and CD4+IFN-γ+Th1 cells in the coculture system (Figs 6A and S4A). When BAY11–7082, the inhibitor of NF-kappa B, was added into this coculture system, the frequencies of Th1 and Th2 cells were both greatly decreased (Figs 6B and S4B), indicating that NF-κB pathway is involved in Th1 and Th2 responses in this fungal infection. Furthermore, upregulation of IL-18Rα expression was detected in CD4+T cells within the co-culture system (Fig 6C). Notably, the expression of IL-18Rα was greatly increased in Th2 cells (nearly 5-fold increase, $P < 0.0001$, Fig 6C), followed by Th1 cells ($P < 0.001$, S4C Fig); however, IL-18Rα expression in Th17 cells remained comparable to the control (S4C Fig).

**IL-18 with IL-2 biased the immune response to Th2 in sporotrichosis caused by *S. globosa***

Previous reports have shown that IL-18, together with IL-2, can induce Th2 cell differentiation in mice [25]. To explore the roles of these cytokines in human Th2 polarization, PBMCs from healthy donors were cultured with IL-18, IL-2, or their combination for 4 days and analyzed by flow cytometry. We found that either IL-18 or IL-2 alone slightly increased the proportion of CD4+IL-4+Th2 cells in PBMCs; however, their combination dramatically enhanced Th2 expansion (Fig 6D), and all the promotive effects were significantly inhibited by BAY11–7082 (Fig 6E). Moreover, the promotion of Th2 cells by IL-18 was dose-dependent, regardless of the presence of IL-2 (Fig 6D).

To verify whether the synergistic effect of IL-18 and IL-2 on Th2 induction naturally occurred in PBMCs exposed to *S. globosa*, we examined IL-2 expression in PBMCs stimulated with heat-killed yeast cells by qPCR and flow cytometry. We found that IL-2 expression was significantly increased at mRNA levels (Fig 7A), and the proportion of CD4+IL-2+T cells was statistically elevated ($P < 0.0001$, Fig 7B). Interestingly, the proportion of CD8+IL-2+ T cells remained unchanged (Fig 7B). Further analysis demonstrated that IL-2 production was predominantly attributed to Th1 cells. In contrast, Th2 and Th17 cells exhibited no significant changes in IL-2 expression during *S. globosa* infection (Fig 7C). Moreover, the concomitant application of IL-2 antibody remarkably augmented the inhibitory impact of anti-IL-18 antibody on Th2 induction in PBMCs-*S. globosa* system (Fig 7D).

We further investigated local expression of IL-2 in sporotrichosis lesions via IHC. Compared with HCs, the number of IL-2+ cells was greatly increased (Fig 7E and 7F), but their distribution was unrelated to disease duration (Fig 7G). Unlike IL-18, IL-2 showed no correlation with IL-4 (Fig 7H), suggesting that IL-2 may not be the dominant factor but just assist IL-18 in Th2 induction.

## Discussion

The results presented here uncover Th2-predominant responses in lesional skin of patients with sporotrichosis caused by *S. globosa*, which is consistent with the changes we previously observed in the circulation of patients [18]. Previous studies have acquired a consensus that a Th1/Th17 profile leads to control of fungal infections in both patients and experimental models [10,13,15,49,50], whereas the involvement of Th2 responses remains controversial. It has been reported that type 2 polarized immunity increases susceptibility to systemic *Candida* infection [51] and progressive cryptococcal disease [52]; and it is associated with elevated fungal burdens and reduced survival in *Cryptococcus neoformans*-infected mice [53], as well as adverse outcomes in *Aspergillus* conidia-infected mice [54,55], due to the suppression of protective Th1 immune responses. Conversely, a beneficial role of early IL-4Rα signaling was found in pulmonary cryptococcosis

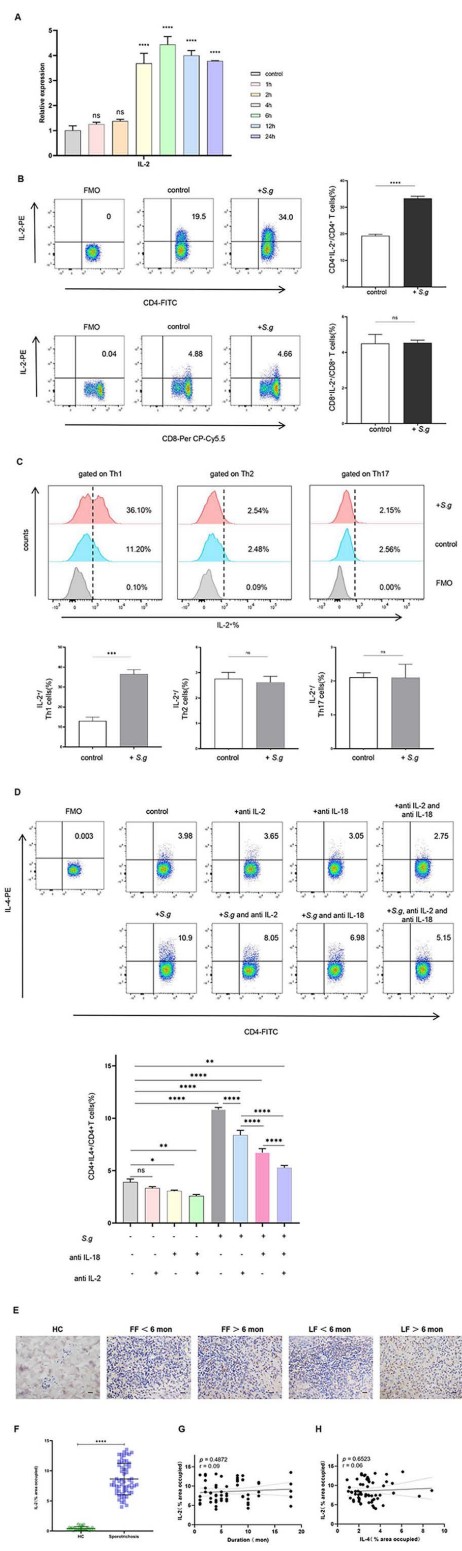

Fig 7.  **The expression of IL-2 in _S. globosa_-infected PBMCs and skin lesions of sporotrichosis .** (A) The mRNA levels of IL-2 in PBMCs after _S. globosa_ stimulation (MOI = 5) at different time points. (B) Flow cytometry analysis of CD4+IL-2+ T cells (above) and CD8+IL-2+ T cells (below) percentages after _S. globosa_ exposure. (C) IL-2 co-expression patterns across Th1 (IFN-γ+), Th2 (IL-4+), and Th17 (IL-17A+) subsets. (D) Effect of neutralizing

IL-18 or/and IL-2 on CD4+IL-4+ Th2 cell proportions during *S. globosa* stimulation, analyzed by flow cytometry. (E) Representative IHC images for IL-2 expression in each group. Original magnification ×400. (F) Statistical analysis of IL-2 dermal distribution area in HCs (n = 5) and sporotrichosis lesional skins (n = 12). (G, H) Correlation analysis between IL-2 dermal expression and disease duration (G) or IL-4 (H) expression (n = 12). The graphs showed a linear fit in patients. The data were analyzed with Pearson correlation analysis. The dotted line demonstrated a 95% confidence interval; r and associated *P* values are shown on the graph. The data shown are expressed as mean ± SD of three independent experiments, and the measured values from individual patients were plotted by dots. *$P < 0.05$, **$P < 0.01$, ***$P < 0.001$, ****$P < 0.0001$, and ns $P \geq 0.05$.

[56]; increases of the Th1 and Th2 cytokines after low-level laser therapy accelerated the healing of the Paracoccidioidomycosis lesions [57]; Th2 and Th17 responses are required for both fungal eradication and prevention of invasion and fungal dissemination in *Candida albicans* (*C. albicans*)-induced allergic airway disease [58]. In our study, we found that only IL-4, not IFN-γ or IL-17A, was strongly positively correlated with the disease duration, suggesting that persistent Th2 responses may contribute to the protracted course of the disease caused by *S. globosa*. Similar to our data, Ji et al. [17] found Th2-type cytokines promoted the progression of sporotrichosis in *S. schenckii*-infected mice. The enhanced Th1 and Th17 responses following laser therapy resulted in improvement of footpad infection caused by *S. globosa* in mice, which indirectly reflects the ineffectiveness of the Th2 response in this disease [59].

Besides, we found that IL-18 played a crucial role in regulating local Th2 responses within the lesions. Previous reports showed IL-18 enhanced protective Th1 immunity against fungal infections such as *Aspergillus* and *C. albicans* [33,34]. Here we observed IL-18 in sporotrichosis lesions was positively correlated with both local expression of IL-4 and IFN-γ. The expansions of Th1 and Th2 cells in PBMCs challenged with *S. globosa* in vitro were both suppressed by IL-18 antibody. These data suggest that IL-18 concurrently regulates these two Th subsets in this specific disease. Although IL-18 has been shown to regulate both Th1 and Th17 cells in psoriasis [46,60–63] and to stimulate both Th1 and Th2 responses in atopic dermatitis [64], such dual regulation has not been observed in other infectious diseases, to the best of our knowledge. Moreover, we found that IL-18 had a stronger correlation with IL-4 (r = 0.70) than with IFN-γ (r = 0.33), indicating IL-18 favors Th2 response in *S. globosa* infection. Coinciding with IL-4, IL-18 expression in the lesions was also positively correlated with disease duration, which further confirms the potent role of IL-18 in promoting Th2 response.

Consistent with previous reports [20,25], IL-2 was confirmed to be the assistant factor to IL-18 in driving Th2 response in PBMCs in vitro. Additionally, we found increased IL-2 expression in lesional skin and in PBMCs exposed to *S. globosa* in vitro, and blocking IL-2 with its antibody further decreased Th2 induction, indicating that IL-2 indeed participates in driving Th2 response with IL-18 in this fungal infection. However, unlike IL-18, IL-2 in the lesions had no correlation with local IL-4 expression and disease duration, which means IL-2 exerts only an auxiliary role to IL-18 in Th2 induction. Intriguingly, our findings revealed that Th1 cells served as the primary source of IL-2 during *S. globosa* infection. We also demonstrated that IL-18 potentiated the Th1 response. It is noteworthy that IL-2 secreted by Th1 cells synergized with IL-18 to promote Th2 cell expansion, challenging the conventional paradigm of reciprocal antagonism between Th1 and Th2 cells [65], highlighting the complexity of immune cell interactions during *S. globosa* infections. Furthermore, BAY 11–7082, a specific inhibitor of IκBα phosphorylation and NF-κB activation [66], greatly reduced the proportion of Th2 cells in *S. globosa*-PBMCs coculture, as well as the induction of Th2 cells by IL-18 and IL-2 in vitro, indicating IL-18 promotes Th2 response via NF-κB signaling pathway during *S. globosa* infection. In addition, a recent report indicates that IL-18 drives secretion of pathogenic cytokines from Th2 cells in atopic dermatitis, with IL-9 upregulating the IL-18R on Th2 cells [28]. We found that IL-18R was elevated in the lesional skin and *S. globosa*-infected CD4+T cells in this study, particularly within the Th2 subset; however, whether IL-9 sensitizes Th2 cells to IL-18 signals in sporotrichosis needs further investigation.

IL-18 is constitutively expressed as pro-IL-18 by hematopoietic and non-hematopoietic cells such as keratinocytes, macrophages and dendritic cells [21,23,64,67,68]. Synthesized as an inactive precursor, it requires processing by the inflammasome/caspase-1 complex for activation and secretion [67,69,70]. The assembly of the NLRP3 inflammasome is essential for the ex vivo release of IL-18 in murine peritoneal macrophages following *S. schenckii* challenge [41]. Upon *S.*

*globosa* infection, an upregulation of NLRP3 and caspase-1 expression levels was observed in the footpads of infected mice [59]. These data indirectly corroborated that the expression of IL-18 was increased during *Sporothrix* infection. Our data showed a significant upregulation of IL-18 expression in sporotrichosis skin lesions compared to normal skin. Notably, both the precursor and mature forms of IL-18 were detected in sporotrichosis lesions, with the mature form being predominant. In contrast, normal skin tissues exclusively expressed pro-IL-18. Furthermore, we observed an increase in the activated form of caspase-1 in PBMCs infected with *S. globosa*, suggesting enhanced inflammasome activation during sporotrichosis infection. These findings align with the Gonçalves' report, in which they detected elevated levels of activated caspase-1 in a BALB/c mouse model following *S. schenckii* infection [40], providing an explanation for the enhanced production of mature IL-18 during *S. globosa* infection. Hu X. et al. demonstrated that IL-18 released through NLRP3-mediated pyroptosis induced Th2 differentiation in allergic rhinitis [71]. Whether a similar mechanism operates in *S. globosa* infection remains to be elucidated and warrants further investigation.

Besides, we found that IL-18 was primarily expressed by DCs and LCs in the dermis of lesional skins. It has been reported that active IL-18 produced by epidermal LCs and skin DCs could promote their migration from skin to draining lymph nodes [63,72–75]. Similarly, we observed the migration of IL-18-secreting LCs from the epidermis to the dermis where lymphocytes were highly populated, suggesting that LCs together with DCs may regulate local T cell responses via IL-18 there. Previous reports demonstrated that FcεRI-activated LCs and DCs contribute to Th2 and Th1 responses, respectively, in vitro, which may result in the shift from Th2 to Th1 response in atopic dermatitis [72]. This finding implies that, within the context of sporotrichosis, LCs and DCs that carry IL-18 could potentially exert differential regulatory impacts on Th1 and Th2 responses. Nevertheless, additional research is required to validate this implication.

Unexpectedly, keratinocytes were not the main source of IL-18 in skin lesions. Despite consensus on the constitutive expression of pro-IL-18 in keratinocytes [23,45], controversy remains regarding its bioactivity. Some studies suggest that keratinocytes produce large amounts of active IL-18 in response to stimuli [47,76,77], while others argue that external stimuli cannot induce active IL-18 production in human keratinocytes [78]. We identified IL-18 by western blot and used recombinant human IL-18 as a size control to distinguish between pro-IL-18 and mature IL-18. Ultimately, we verified that the IL-18 synthesized by keratinocytes was the precursor form, consistent with the findings reported by Mee JB. et al. [78].

While our investigation has yielded novel insights into the immune-pathological mechanisms of this disease, it lacks in vivo validation through preclinical models to directly confirm IL-18/Th2-mediated effects on disease outcome. This is partly restricted by the disparities in immune responses to this fungus between mouse and humans as we observed Th17 dominant responses in murine *S. globosa* footpad infection model (S5 Fig) instead of Th2 skewed responses in patients. However, our previous work has strongly indicated that *S. globosa* melanin promotes the polarization of both murine and human macrophages towards an alternatively activated (M2) phenotype as manifested by their suppressed abilities of antigen presentation and phagocytosis, and reduced productions of proinflammatory mediators (NO, TNF-α, IL-6, and IL-1β) [79,80]. Given that Th2 skewed microenvironments are known to promote M2 macrophage polarization [81], and considering the well-established role of M2 macrophages in fungal immune evasion [79,80,82,83], we hypothesize that IL-18/Th2-mediated signaling converges on M2 macrophage polarization, creating an immune niche conducive to *S. globosa* persistence and disease chronicity. This hypothesis is consistent with our current findings, but still needs direct experimental verification in future research.

Overall, we demonstrate for the first time that skin lesions of sporotrichosis patients caused by *S. globosa* are characterized by a mixed Th1/Th2/Th17 response with a predominant Th2 profile. Furthermore, IL-18 simultaneously modulates both Th1 and Th2 responses but favors the Th2 response, which is enhanced by IL-2 mainly produced by Th1 cells. This Th2-skewed immune response significantly contributes to disease progression. Further in-depth research into how IL-18 precisely modulates Th1 and Th2 cells within a complex cytokine milieu will better elucidate the immune mechanisms underlying the disease and provide more targeted therapeutic strategies for clinical management.

## Supporting information

**S1 Appendix. Supporting materials and methods.**
(DOCX)

**S1 Table. Clinical characteristics of patients enrolled.**
(DOCX)

**S2 Table. Antibodies and cytokines used in cell culture, IHC, mIHC, and WB.**
(DOCX)

**S3 Table. Primers for qPCR.**
(DOCX)

**S1 Data. Data that underlies this paper.**
(XLS)

**S1 Fig. The expression of caspase-1 in skin lesions of patients with sporotrichosis caused by *S. globosa*.** (A) qPCR analysis for mRNA expression of caspase-1 in HC skins and sporotrichosis lesional skins. (B) Representative IHC images for caspase-1 expression in each group. Original magnification ×400. (C) Statistical analysis of caspase-1 distribution area in the dermis of HC skin and sporotrichosis lesional skin (above); with comparisons between LD and SD (middle); between FF and LF (below). (D) Correlation between dermal caspase-1 expression and disease duration. (E) Statistical analysis of caspase-1 distribution area and average optical density (AOD) in the epidermis of HC skins and sporotrichosis lesional skins. Sample sizes: HC: n = 5, lesion: n = 20, n = 5 per subgroup. The data shown are expressed as mean ± SD of three independent experiments, and the measured values from individual patients were plotted by dots. The graphs showed a linear fit for patients. The data were analyzed with Pearson correlation analysis. The dotted line demonstrated a 95% confidence interval; Pearson's correlation coefficient (r) and associated $P$ values are shown on the graph. $*P < 0.05$, $**P < 0.01$, $***P < 0.001$, $****P < 0.0001$, and ns $P > 0.05$.
(TIF)

**S2 Fig. Identified the form of IL-18 protein produced by human primary keratinocytes and PBMCs.** Western blot analysis of IL-18 expression in keratinocytes and PBMCs after *S. globosa* stimulation. Lane 1: recombinant human IL-18; lane 2: human primary keratinocytes lysates; lane 3: PBMCs lysates. The lysates were from three independent experiments.
(TIF)

**S3 Fig. The expression of caspase-1 in PBMCs after exposure to *S. globosa* in vitro.** (A) The mRNA levels of caspase-1 in PBMCs after *S. globosa* stimulation (MOI = 5) at different time points. (B) Western blot analysis of cleaved caspase-1 expression in PBMCs after *S. globosa* stimulation (MOI = 5) at different time points (left). Statistical analysis of relative protein expression was quantified with ImageJ software (right). The data shown are expressed as mean ± SD of three independent experiments. $*P < 0.05$, $**P < 0.01$, $***P < 0.001$, $****P < 0.0001$, and ns $P > 0.05$.
(TIF)

**S4 Fig. The effect of IL-18 on Th1 and Th17 responses during *S. globosa* infection.** (A, B) Effect of neutralizing IL-18 (A) and blocking NF-κB (B) on CD4+IFN-γ+ Th1 and CD4+IL-17A+ Th17 immune responses during *S. globosa* stimulation, analyzed by flow cytometry. (C) The expression of IL-18Rα in CD4+IFN-γ+ Th1 and CD4+IL-17A+ Th17 cells. The data shown are expressed as mean ± SD of three independent experiments. $*P < 0.05$, $**P < 0.01$, $***P < 0.001$, $****P < 0.0001$ and ns $P \geq 0.05$.
(TIF)

**S5 Fig. Kinetics of Th1/Th2/Th17 responses in *S. globosa*-infected footpads.** (A-D) Cytokine profiles (IFN-γ/IL-4/IL-17A) quantified by immunohistochemistry at 1, 2, 4, and 5 weeks post-injection of $1 \times 10^7$ heat-killed *S. globosa* yeast cells (6–8-week-old BALB/c mice; PBS-injected controls). Individual data points from all mice are displayed, with results presented as mean ± SD. *$P < 0.05$, **$P < 0.01$, ***$P < 0.001$, ****$P < 0.0001$ and ns $P \geq 0.05$. Con, control. (TIFF)

## Acknowledgments

We would like to thank Dr. Xiuzhu Gao and Tingshuang Xu in the core facility of the First Hospital of Jilin University for their essential support and services of the flow cytometry, which significantly contributed to our research.

## Author contributions

**Conceptualization:** Henan Si, Yu Zhen, Shanshan Li.

**Data curation:** Henan Si, Yan Cui, Jianjiao Zu, Ying Shi, Yang Song.

**Formal analysis:** Henan Si, Yan Cui.

**Funding acquisition:** Yang Song, Yu Zhen, Shanshan Li.

**Methodology:** Henan Si, Yan Cui, Yang Song, Yu Zhen, Shanshan Li.

**Supervision:** Yu Zhen, Shanshan Li.

**Writing – original draft:** Henan Si, Yu Zhen.

**Writing – review & editing:** Yu Zhen, Shanshan Li.

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
