## [Decision Letter · Decision Letter 0]

Dear Dr. Si,

Please submit your revised manuscript within 30 days Jul 11 2025 11:59PM. If you will need more time than this to complete your revisions, please reply to this message or contact the journal office at plosntds@plos.org. Please include the following items when submitting your revised manuscript:

Response to Reviewers
Revised Manuscript with Track Changes
Manuscript

We look forward to receiving your revised manuscript.

Shaden Kamhawi

co-Editor-in-Chief

Paul Brindley

co-Editor-in-Chief

**Journal Requirements:**
**Reviewers' comments:**

**Key Review Criteria Required for Acceptance?**

**Methods**

-Are the objectives of the study clearly articulated with a clear testable hypothesis stated?

-Is the study design appropriate to address the stated objectives?

-Is the population clearly described and appropriate for the hypothesis being tested?

-Is the sample size sufficient to ensure adequate power to address the hypothesis being tested?

-Were correct statistical analysis used to support conclusions?

-Are there concerns about ethical or regulatory requirements being met?

Reviewer #1: -Are the objectives of the study clearly articulated with a clear testable hypothesis stated?: The objectives are clear and related to the hypothesis proposed.

-Is the study design appropriate to address the stated objectives?: The study design is consistent with the objectives set.

-Is the population clearly described and appropriate for the hypothesis being tested?: Yes, the study population and the number of observations made have been defined.

-Is the sample size sufficient to ensure adequate power to address the hypothesis being tested?: The sample size is adequate and allows the inferences made by the authors to be drawn.

-Were correct statistical analysis used to support conclusions?: Statistical analysis supports the conclusions drawn.

-Are there concerns about ethical or regulatory requirements being met?: Yes.

Reviewer #2: Are the objectives of the study clearly articulated with a clear testable hypothesis stated?

Yes

-Is the study design appropriate to address the stated objectives?

Yes

-Is the population clearly described and appropriate for the hypothesis being tested?

Yes

-Is the sample size sufficient to ensure adequate power to address the hypothesis being tested?

Uncertainty

-Were correct statistical analysis used to support conclusions?

Yes

-Are there concerns about ethical or regulatory requirements being met?

Yes

**Results**

-Does the analysis presented match the analysis plan?

-Are the results clearly and completely presented?

-Are the figures (Tables, Images) of sufficient quality for clarity?

Reviewer #1: -Does the analysis presented match the analysis plan?: The analysis performed corresponds to what was planned in the methodology.

-Are the results clearly and completely presented?: The results clarify the hypothesis and allow correlation with findings in previous studies.

-Are the figures (Tables, Images) of sufficient quality for clarity?: Yes.

Reviewer #2: Does the analysis presented match the analysis plan?

Yes

-Are the results clearly and completely presented?

Yes

-Are the figures (Tables, Images) of sufficient quality for clarity?

Yes

**Conclusions**

-Are the conclusions supported by the data presented?

-Are the limitations of analysis clearly described?

-Do the authors discuss how these data can be helpful to advance our understanding of the topic under study?

-Is public health relevance addressed?

Reviewer #1: -Are the conclusions supported by the data presented?: Yes.

-Are the limitations of analysis clearly described?: Yes

-Do the authors discuss how these data can be helpful to advance our understanding of the topic under study?: Yes.

-Is public health relevance addressed?: This study highlights the importance of IL-18 with a dual mechanism of action related to the regulation of Th1/Th2/Th17 immunity with a predominance of Th2 and which, through complex immunological mechanisms, allows local control of the disease. Although the study allows hypotheses to be established that may be relevant in future clinical and therapeutic studies, it is not yet possible to determine the precise relevance of the finding in public health.

Reviewer #2: Are the conclusions supported by the data presented?

Yes

-Are the limitations of analysis clearly described?

Yes

-Do the authors discuss how these data can be helpful to advance our understanding of the topic under study?

Yes

-Is public health relevance addressed?

Yes

**Editorial and Data Presentation Modifications?**

Reviewer #1: The authors clearly outline the local and systemic immune mechanisms that influence the chronicity of sporotrichosis. I am satisfied with the experiments conducted, which have clarified that many of these mechanisms are similar (if not identical) to those reported in other research involving other species of Sporothrix.

Reviewer #2: Minor Revision

<!--StartFragmentThe length of "Introduction and Methods parts" is a bit too long, which should be polished. <!--EndFragment

Tables 2 and 3: move these to supplemental material as they don't need to be in main text. 

**Summary and General Comments**

Reviewer #1: The authors complement their previous research (Front Immunol. 2020 Nov 13;11:570888) with the results presented in this study, which establish the dual functions of IL-18 as a local promoter of the Th1/Th2/Th17-Treg axis and the superiority of Th2 as the final mechanism influencing the chronicity and low aggressiveness of the disease locally through mechanisms mediated by NLRP3-Caspase 1.

The study itself is revealing in terms of the local mechanisms controlling the progression of sporotrichosis, although it does not allow conclusions to be drawn about the importance of other inflammatory mediators such as IL-4, IL-12, and IL-23 and the inflammasome in general in local control, and not only IL-18 as the main actor in these mechanisms. Nevertheless, it represents a major advance in the understanding of the immunopathogenesis

Reviewer #2: Minor Revision

PLOS authors have the option to publish the peer review history of their article (what does this mean? ). If published, this will include your full peer review and any attached files.

**Do you want your identity to be public for this peer review?** For information about this choice, including consent withdrawal, please see our Privacy Policy .

Reviewer #1: **Yes: ** Andrés Tirado-Sánchez

Reviewer #2: No

**Figure resubmission:****Reproducibility:** To enhance the reproducibility of your results, we recommend that authors of applicable studies deposit laboratory protocols in protocols.io, where a protocol can be assigned its own identifier (DOI) such that it can be cited independently in the future. Additionally, PLOS ONE offers an option to publish peer-reviewed clinical study protocols. Read more information on sharing protocols at https://plos.org/protocols?utm_medium=editorial-email&utm_source=authorletters&utm_campaign=protocols

---

## [Editor Report · Decision Letter 1]

Dear Dr Si,

Thank you for the rigorous and complete response to the points raised in the review of your work. We are pleased to inform you that your manuscript 'IL-18 favors Th2 responses in sporotrichosis caused by Sporothrix globosa, prolonging the course of the disease' has been provisionally accepted for publication in PLOS Neglected Tropical Diseases.

Best regards,

Joshua Nosanchuk, MD

Section Editor

Shaden Kamhawi

co-Editor-in-Chief

Paul Brindley

co-Editor-in-Chief

---

## [Editor Report · Acceptance letter]

Dear Dr Si,

We are delighted to inform you that your manuscript, "IL-18 favors Th2 responses in sporotrichosis caused by Sporothrix globosa, prolonging the course of the disease," has been formally accepted for publication in PLOS Neglected Tropical Diseases.

Best regards,

Shaden Kamhawi

co-Editor-in-Chief

Paul Brindley

co-Editor-in-Chief
